# A Novel Method for the Background Signal Correction in SP-ICP-MS Analysis of the Sizes of Titanium Dioxide Nanoparticles in Cosmetic Samples

**DOI:** 10.3390/molecules27227748

**Published:** 2022-11-10

**Authors:** Zaual A. Temerdashev, Olga A. Galitskaya, Mikhail A. Bolshov

**Affiliations:** 1Department of Analytical Chemistry, Faculty of Chemistry and High Technologies, Kuban State University, 350040 Krasnodar, Russia; 2Institute of Spectroscopy, Russian Academy of Sciences, 108840 Moscow, Russia

**Keywords:** titanium dioxide nanoparticles, single particle ICP-MS, background signal correction, cosmetic products

## Abstract

We discuss the features involved in determining the titanium dioxide nanoparticle (TiO_2_NP) sizes in cosmetic samples via single particle inductively coupled plasma mass spectrometry (SP-ICP-MS) in the millisecond-time resolution mode, and methods for considering the background signal. In the SP-ICP-MS determination of TiO_2_NPs in cosmetics, the background signal was recorded in each dwell time interval due to the signal of the Ti dissolved form in deionized water, and the background signal of the cosmetic matrix was compensated by dilution. A correction procedure for the frequency and intensity of the background signal is proposed, which differs from the known procedures due to its correction by the standard deviation above the background signal. Background signals were removed from the sample signal distribution using the deionized water signal distribution. Data processing was carried out using Microsoft Office Excel and SPCal software. The distributions of NP signals in cosmetic product samples were studied in the dwell time range of 4–20 ms. The limit of detection of the NP size (LOD_size_) with the proposed background signal correction procedure was 71 nm. For the studied samples, the LOD_size_ did not depend on the threshold of the background signal and was determined by the sensitivity of the mass spectrometer.

## 1. Introduction

Titanium dioxide (TiO_2_) is widely used in the cosmetic industry due to its pigment properties and effective protection from ultraviolet (UV) radiation [1]. Largely, the TiO_2_ particle sizes determine the degree of protection from UV radiation. The most effective UV radiation blockers are nanoparticles (NPs) in a size range of up to 100 nm [1]. TiO_2_ NPs used in the cosmetic industry are safe for direct contact with human skin [2]. On the other hand, other routes of exposure to toxicological and carcinogenic effects of TiO_2_ NPs exist [3]. The wide use of cosmetic products containing TiO_2_ NPs results in various scenarios of NP interactions with the environment, for example, their release upon contact with water, changes under the influence of UV radiation, and others [4,5,6].

A comprehensive assessment of the potential risks associated with TiO_2_ NPs requires a thorough study of NPs in commercial cosmetic products as the sources of their release into the environment. The sizes of TiO_2_ NPs are important parameters that require mandatory control [7].

The sizes of TiO_2_ NPs in cosmetic samples are determined by electron microscopy [8,9,10,11], dynamic light scattering (DLS) [9,10,12,13], X-ray diffraction spectrometry [8,9], field flow fractionation (FFF) with various detection methods [13,14,15,16,17,18], single particle inductively coupled plasma mass spectrometry (SP-ICP-MS) [8,12,13,14,17,19,20,21], and others. The complex composition of the cosmetic matrix makes the identification and determination of TiO_2_ NP sizes difficult. Electron microscopy, a reference method used for the determination of NP sizes, requires complex procedures for separating TiO_2_ NPs from the matrix for analysis under high vacuum conditions. Light scattering methods do not distinguish between NPs of different types and do not represent their real sizes [9]. An X-ray diffraction analysis is insensitive to NPs with sizes exceeding 100 nm [8]. Field flow fractionation effectively separates TiO_2_ NPs into fractions with narrower size distributions, but requires a long analysis time and careful choosing of elution conditions [14,18].

SP-ICP-MS is an optimal method for determining the sizes of TiO_2_ NPs in a multicomponent cosmetic matrix [1]. SP-ICP-MS allows the identification and estimation of TiO_2_ NP sizes with high throughput and satisfactory reproducibility [8,12,13,14,17,19,20,21]. The method allows recording the signal intensity of each single TiO_2_ NP successively introduced into the spectrometer. The signal intensity is proportional to the NP mass after preliminary calibration of the system [22,23]. The papers devoted to the analysis of TiO_2_ NPs in cosmetic products [8,12,13,19] demonstrate the efficiency of SP-ICP-MS as an independent method for determining the sizes of NPs and the possibility of their direct connection with FFF systems [14,17]. A satisfactory correlation between the results of SP-ICP-MS and other methods including electron microscopy has been reported [8]. The fields of application of SP-ICP-MS for the determination of TiO_2_ NPs have expanded, and the possibility of simultaneous determination of the size and number of NPs together with the total titanium content in the sample makes the analysis more informative [17,19,24]. The SP-ICP-MS method can be used as a reliable tool for the rapid screening of TiO_2_ NPs in cosmetic products and implemented in analytical laboratories and industrial production for quality control and risk assessment [12,13,20,21].

On the other hand, SP-ICP-MS has a relatively high limit of detection of the NP size (LOD_size_), which limits its use for small TiO_2_ NPs [25,26]. The signals of single NPs in SP-ICP-MS are detected as high-intensity peaks above the continuous background signal of the dissolved element. However, low-intensity signals of small NPs can be shielded by multiple background signals of similar intensities, which determine the LOD_size_. The LOD_size_ depends on the detected isotopes, sample matrix, operational parameters, and other factors, and depends on the ability to distinguish the NP pulse signals from the background signals, as well as on the detector’s sensitivity to a specific element [27]. To distinguish NP signals from background signals, a threshold value of the sum of the mean and several standard deviations (σ) of the background signal is usually used [19,20,22]. The algorithms of cluster analysis [28,29] and signal deconvolution [30] have also been proposed for the discrimination of NP signals from the background. According to [4,27], the LOD_size_ of TiO_2_ NPs was in the range of 95–130 nm, depending on the detected isotope and the threshold value of the standard deviation of the background signal. The LOD_size_ can be improved by using a collision cell or ms/MS technology in combination with reaction cell gases to reduce matrix interference. The LOD_size_ of TiO_2_ NPs of 37–55 nm was achieved by using a collision cell [24]. A shorter microsecond dwell time reduced the background signal and lowered the LOD_size_ of TiO_2_ NPs to 30–35 [12,13,19] and 65–70 nm [31] for isotopes ^48^Ti and ^47^Ti, respectively. In this case, the use of specialized software for data acquisition and processing is required, which allows the detection of a NP as a transient peak over several dwell times. The software allows automatic summarizing of the fragmented NP signal and it gives the total NP intensity. To date, open-source platforms are available for SP-ICP-MS analyses [32].

In this paper, we studied the features involved in the millisecond SP-ICP-MS determination of the sizes of TiO_2_ NPs in cosmetic products and considered possible background signal estimations. Data processing of the SP-ICP-MS analysis was performed by Microsoft Office Excel without the use of specialized commercial software.

## 2. Results and Discussion

### 2.1. Background Signal in SP-ICP-MS Analysis of TiO_2_ NPs

The intensity of the background signal at the selected *m*/*z* (48 for Ti and 107 for Ag) is proportional to the concentration of the dissolved element, interferents in the sample matrix, and reagents used in the sample preparation. Besides TiO_2_ NPs, cosmetic products may contain Ti in dissolved form. In addition, cosmetic products often contain calcium compounds, and sulfur compounds are also found. ^48^Ca (abundance 0.19%) and ^16^O^32^S represent interfering influences in the analysis of ^48^Ti, increasing the continuous signal at *m*/*z* 48. Strict compliance with the conditions for registration of each single TiO_2_ NP separately from other NPs requires dilution of the cosmetic matrix suspension with deionized water to achieve optimal concentrations of NPs in the solvent. The effect of the suspension dilution on the background signal at *m*/*z* 48 was studied for samples 1 and 2 with typical cosmetic matrices without TiO_2_ NPs. The sample suspensions were successively diluted 10, 100, and 1000 times with deionized water. The signal intensity monitoring at *m*/*z* 48 showed that dilution of the sample suspension by 1000 times resulted in reducing the content of dissolved Ti and interferents in the cosmetic matrix and lowering a signal to 15 counts, which corresponds to the background signal of deionized water (Appendix A). The obtained result shows that, in the case of a significant dilution of the cosmetic matrix, the LOD_size_ of TiO_2_ NPs is mostly limited by the purity of deionized water.

Then, the background signal was studied in a time-resolved mode. The signal at *m*/*z* 48 was monitored in the dwell time range of 0.1–20 ms at constant values of other measurement conditions. The mean intensity of the background signal was dependent on dwell time (Appendix A). An increase in dwell time resulted in a greater signal accumulation and a linear increase in its mean intensity (Appendix A). The intensity at the distribution maximum was close to the mean intensity of the background signal (Appendix A). The signal intensities at the wings of the ranges had a significantly lower frequency (except for signals with dwell times less than 1 ms, for which the frequency of low-intensity signals is maximum).

With an increase in dwell time, an increase in signal fluctuations expressed in a broadening of the signal distribution was observed (Appendix A). For example, the ranges of signal intensities at 0.1, 1, and 10 ms dwell times were 0–5, 0–16, and 28–70 counts, respectively (Appendix A). At the same time, the width of the range followed an exponential dependence. The largest growth of the distribution width was observed within the variation of dwell times up to 5 ms, and then the width decreased (Appendix A). The change in the mean intensity (the position of the distribution maximum on the intensity axis) remained proportional throughout the tested dwell time interval (Figure 1a,b).

The mean signal intensity in this range of dwell times practically did not exceed the intensity at the distribution maximum but did not show the real position of the right wing of the distribution along the intensity axis. The assessment of the background signal level using the nσ-threshold (a threshold value of several standard deviations above the mean background signal) allowed for characterizing the background signal more correctly (Appendix A). 

On the other hand, the increase in dwell time resulted in a broadening of the distribution. The standard deviation value of the signal σ was expected to increase exponentially with an increase in the width of the distribution, which led to an increase in the threshold of the background signal calculated by 3σ, 5σ, and 10σ criteria (Appendix A). The threshold of the background signal calculated by 3σ at dwell time up to 7 ms left the high–intensity part of background signals in the distribution; within the dwell time interval from 7 to 10 ms background signals sufficiently corresponded to the real maximum intensity of the background signal; above 10 ms–exceeded it. 

The threshold calculated by 5σ correctly characterized the background signal in the dwell time range of up to 3 ms and was overestimated at longer dwell times. The threshold of the background signal calculated by 10σ in the entire dwell time range was excessively overestimated relative to the real intensity. The selection of the threshold criteria for the discrimination of the background signals from the NP signals remains on the researcher, but it must be reasoned. Background signals should not be taken as false positive NP signals, and low-intensity NP signals should not be taken as false negatives when processing the results.

At dwell times of 1 ms or less, windows with an intensity of 0 counts were observed, i.e., the background signal was not registered at a short dwell time (Appendix A). In this case, an increase in the background signal with an increase in dwell time theoretically leads to a decrease in the signal-to-noise ratio, while the signal depending on the size of NP remains unchanged, and the background signal increases. This leads to the loss of signals of small NPs in the array of background signals and a decrease in the LOD_size_ of TiO_2_ NPs. At dwell times exceeding 1 ms, the background signal was detected in each window (Appendix A). Under these conditions, the detected signal of TiO_2_ NPs corresponds to the sum of the intensities of the background and NP signals.

### 2.2. Signals of Single TiO_2_ NPs

The signal distributions of TiO_2_ NPs in cosmetic products 3, 4, and 5 were studied.

To determine the presence of NPs in the samples, tests were carried out on the IG-1000 nanoparticle size analyzer (Shimadzu, Japan). The obtained particle size distributions (hydrodynamic diameter) for samples 3, 4, and 5 were in the ranges 174–575, 174–437, 229–575 nm, respectively (Appendix A). Note that the capabilities of the IG-1000 do not allow identifying the type of NPs and give a general size distribution for all nanoscale objects in the sample. The measured NP sizes include a hydrate shell and give an overestimated result relative to their true size, and the pronounced polydispersity leads to a shift in the size distribution in a larger direction [12].

The SP-ICP-MS distributions of the sample and background signals in sample 3 obtained under the same conditions at 10 ms dwell time are presented in Figure 1. Both distributions have monomodal profiles with the identical position of distribution maxima on the intensity axis. As can be seen, the second maximum of the signal distribution corresponding to TiO_2_ nanoparticles is absent. At the same time, the distribution of the sample signals has peaks with intensities exceeding the upper limit of the background signal intensity of 48 counts, which is clearly seen in the enlarged scale of the ordinate axis “Frequency” (Figure 1b). This confirms the presence of TiO_2_ NPs with intensities exceeding the background level in the sample.

The position of the maxima of the background and sample signal distributions on the intensity axis coincides. This shows that the available maximum of the monomodal sample signal distribution cannot characterize TiO_2_ NP size. The background signal makes a significant contribution to the distribution of sample signals.

Further, the sample dilution was studied in the range of 5000 to 50,000 times under the same measurement conditions. An increase in the dilution factor of sample 5 resulted in a decrease in the number of high-intensity signals, which characterize TiO_2_ NPs (Figure 2). At the same time, the distribution of background signals remained unchanged and had a constant maximum, while the number of TiO_2_ NPs decreased proportionally to the degree of the sample dilution.

The signal distribution of sample 3 containing TiO_2_ NPs was also studied at 100 ms dwell time. The sample was studied with excessive dilution, excluding the simultaneous detection of two or more NPs. The total measurement time was 10 min, which allowed for obtaining a sufficient number of NP signals. On the time-resolved spectra of sample 3 at 10 and 100 ms dwell times, the peaks of NPs above the background signal are similar and, with rare exceptions, exceed the background signal by approximately 200 counts in both cases (Figure 3). At the same time, the intensity of the background signal increases from 100 to 500 counts (the continuous color area in the lower part of the spectrum in Figure 3a,b) with an increase in dwell time from 10 to 100 ms. There was no loss of high-intensity NP signals in the array of background signals. 

This confirms the conclusion that dwell times longer than 1 ms provide the detection of background signals in each single dwell time window. The measured signal during the registration of TiO_2_ NPs is represented by the cumulative signal of the background and the particle. When dwell time increases, the measured particle signal increases proportionally to the increase in the background signal.

The increase in the upper level of the background signal was approximately 400 counts, which allowed for predicting the same increase in the total intensity of the particle and background signals. The signal distributions of sample 3 at 10 and 100 ms dwell times were characterized by a similar distribution of NP signals at the right wing and were shifted relative to each other by approximately 400 counts along the intensity axis (Figure 3c). The significant impact of the background signal on the position of the distribution on the intensity axis excludes the possibility of using the intensities of NP peaks to calculate particle sizes without background signal correction.

### 2.3. Background Signal Correction for the Monomodal Distribution of Sample Signals

In the case when the distribution of NP signals is inseparable from the background signal distribution, the possibility of visual determination of the background signal level is practically excluded. The use of σ-criteria allowed for calculating the signal threshold level, which cuts off all signals with intensities lower than the calculated background intensity (red line in Figure 4a). However, background signals with maximum intensity in the range of 40–50 counts at the right wing of the distribution had low frequency, whereas, in the distributions of samples, signals in the same intensity range had a high frequency and were partially signals of TiO_2_ NPs (Figure 4a). In this case, the background signal correction using the σ-threshold leads either to an overestimation of the number of NPs due to false-positive background signals, when the background signal threshold is insufficient to completely cut off the background signals (3σ in Figure 4b), or to an underestimation of the number of NPs due to the cut-off of NP signals with an intensity similar to that of the background signal at the right wing of the distribution (5σ and 10σ in Figure 4c,d). In both cases, the background signal correction using the σ-threshold turned out to be ambiguous and led to an unsatisfactory result.

An increase in the difference between the minimum and maximum of the background signal intensity led to an increase in the standard deviation σ and, consequently, to an increase in the background signal threshold (Appendix A). At dwell times exceeding 10 ms, the calculated 3σ-threshold of the background signal exceeded the real intensity at the right wing of the background signal distribution. In this case, the loss of NP signals is unavoidable. 

Signal intensity data of cosmetic product samples were also processed using the SPCal software [32], which allows automatic calculation of the background signal using Poisson or Gaussian distributions, depending on the data. Appendix A contains calculated LOD values for sample 3 at a dwell time in the range of 1–20 ms. A correlation was observed between LOD values with increasing dwell time. The LOD values were quite close to the LOD values calculated from the deionized water signal distributions (Appendix A). The TiO_2_ NP number above the calculated background signal is small (32–131 particles), and most of the registered NPs have been removed from the distribution (Appendix A). 

In the case of TiO_2_ NP signal distribution, when the TiO_2_ NP signal distribution is inseparable from the background signal distribution, the traditional correction of the background signal using the σ-threshold is redundant and requires the search for other approaches. 

The background and sample signal distributions are identical under the same operational parameters, excluding the areas corresponding to the overlap of NP signals at the wing of the background distribution (Figure 1). It can be expected that subtracting the background signal distribution from the sample signal distribution according to Equation (1) will allow calculating the number of signals not corresponding to background signals
(1)$$N_{I}^{s}-N_{I}^{\mathrm{bgd}}=N_{I}^{s-\mathrm{bgd}},$$
where $N_{I}^{s}$ and $N_{I}^{\mathrm{bgd}}$ are the number of signals of the sample and background with intensity I.

Subtracting the number of signals of each intensity for two signal distributions (Figure 5a) allowed us to construct a new “sample–bgd” distribution, and its part is located in the negative range on the ordinate axis “Frequency” (Figure 5b). The number of signals in the negative and positive ranges was equal, and their values were modulo equivalent to the number of detected particles ($N_{\mathrm{NP}}$) calculated for this sample by Equation (2).
(2)$$N_{\mathrm{NP}}=\sum^{}\left|N_{I}^{s-\mathrm{bgd}}\right|,$$
where $N_{I}^{s-\mathrm{bgd}}$ is the difference between the number of the sample and background signals with intensity I in the negative range on the ordinate axis “Frequency”.

The number of background signals $N_{\mathrm{bgd}}$ was calculated by Equation (3).
(3)$$N_{\mathrm{bgd}}=N-N_{\mathrm{NP}},$$
where *N* is the total number of signals (dwell time windows) equal to the sample and background distributions.

With the initial equality of the total number of dwell time windows *N* on the sample and background spectra, the NP signal in one dwell time window on the sample spectrum conditionally replaces one dwell time window on the background signal spectrum. Respectively, the number of background signals in the signal distribution of the sample decreases by the number of NPs in this sample, which is not taken into account in Equation (1).

For reliable correction of the background signal, the number of background distribution signals $N_{I}^{\mathrm{bgd}}$ was recalculated considering the number of NP signals in the sample, i.e., reduced by their average number according to Equation (4):(4)$$N{\mathrm{corr}}_{I}^{\mathrm{bgd}}=\;N_{I}^{\mathrm{bgd}}\times \frac{N_{\mathrm{bgd}}}{N},$$
where $N{\mathrm{corr}}_{I}^{\mathrm{bgd}}$ is the corrected number of background signals with intensity I.

Considering the assumptions made, subtracting the corrected distribution of background signals from the distribution of sample signals according to Equation (5) will result in obtaining the corrected distribution of sample signals “sample–bgd_corr_”, which will contain only NP signals.
(5)$$N_{I}^{s}-N{\mathrm{corr}}_{I}^{\mathrm{bgd}}=N{\mathrm{corr}}_{I}^{s-\mathrm{bgd}},$$

However, in the case of a real sample analysis with a polydisperse distribution of TiO_2_ NP sizes, including those with NP sizes below the LOD_size_ defined by the background signal, the number of background signals with certain intensities will exceed the number of NP signals. The corrected distribution of the background signal does not permit accounting for the signals of small NPs with intensities equal to the intensities of background signals at the upper limit. The calculated number of background signals $N_{\mathrm{bgd}}$ turns out to be overestimated at the left wing of the distribution in the low-intensity range. This confirms the assumption about the inaccuracy of the calculation caused by the small NPs. As a result, the new corrected “sample–bgd_corr_” distribution has a number of signals in the negative range, but significantly less than before the correction (Figure 5c).

At this stage, it is possible to visually determine the threshold of the background signal—the signals of NPs and backgrounds are differentiated, and part of the background signals with an intensity higher than that at the point of transition of the distribution from the negative frequency range to the positive is extracted from the signal distribution of the sample (Figure 5c). The optimal option, in this case, was to consider the mean intensity at the maximum of the background signal distribution as a background signal, which corresponded to the intensity at the point of transition. The use of the mean intensity of the background signal to correct the distribution by the σ-threshold previously proved insufficient at removing high-intensity background signals. However, due to the preliminary separation of the background and NP signals, in this case, there is no need to use background signal intensities exceeding the mean value. Subtracting the mean intensity of the background signal from the intensity of each sample peak allowed us to obtain the final distribution of the “sample_corr_”, corrected for the number of background signals and their intensity (Figure 5d).

### 2.4. Determination of TiO_2_ NP Size in Cosmetic Product Samples

The distributions of TiO_2_ NP signals in samples 3–6 were studied in the dwell time range of 4–20 ms. In these experiments, we selected a dwell time window sufficient to detect the signal intensity of an entire single particle. The distributions of TiO_2_ NP signals in the samples were obtained under identical conditions, followed by the same data processing. The conversion of intensity data into NP sizes was carried out considering the transport efficiency of the system and the calibration curve [23].

Data processing of the results of the analysis of cosmetic products with TiO_2_ NPs without background signal correction showed an increase in the mean particle size at the distribution maximum with an increase in dwell time (Table 1, Appendix A). The shift of the distribution maximum along the intensity axis as well as the shift along the particle size axis after calibration were previously shown to be due to an increase in the intensity of the background signal with an increase in dwell time. At the same time, the NP signals at the right wing of the distributions (Appendix A) remain unchanged and shift along the intensity (particle size) axis proportionally to the change in the background signal since they represent peaks with the additive intensity of the background and TiO_2_ NPs.

The absence of background signal correction invariably caused a difference in the estimation of the particle sizes for one sample (Table 1). Background signal correction using the σ-threshold became ambiguous for the monomodal distribution of the background and TiO_2_ NP signals.

Correction of the signal distributions of samples 3–6 by the frequency and intensity of the background signals made it possible to significantly compensate for the shift of the distribution along the particle size axis. Appendix A shows the change in the position of the signal distributions of the samples on the particle size axis before and after background signal correction at different dwell times. The uncorrected signal distributions of samples 3–6 shifted significantly along the particle size axis with increasing dwell time, while the position of the corrected particle distributions remained relatively constant. For example, for sample 4, the position of the uncorrected maximum signal distribution shifted within 146–248 nm with an increase in dwell time from 4 to 20 ms, while after correction it was in the range of 122–146 nm (Table 1). The correction procedure allowed compensating for the background signal, which further did not affect the shift in the distribution of NP signals (Appendix A). It can be assumed that the shift of the maximum of the NP size distribution with an increase in dwell time for each sample is caused by the splitting of the NP signals because of the insufficient length of the short dwell times. The position of the distribution maximum for samples 3 and 4 remained constant at dwell times longer than 16 ms and slightly changed for sample 5. It can be assumed that dwell times longer than 16 ms are sufficient to register an entire NP, while the variation of the mean size of TiO_2_ NPs after the background signal correction under optimal conditions did not exceed 4 nm for samples 3–5 (Table 1).

Correction of the distributions to the frequency of the background signal also allowed determining the number of single TiO_2_ NPs detected for each sample. However, the number of TiO_2_ NPs varied with increasing dwell time (Table 1), and the change in the distribution function of TiO_2_ NPs is visible for all samples (Appendix A). In all cases, the increase in dwell time from 4 to 16 ms caused a decrease in the number of TiO_2_ NPs at the distribution maximum and also the shift of the maximum to large size values. Such a change is similar for all analyzed samples and is caused, apparently, by the peculiarities of detecting single NPs. It is known that an increase in the dwell time window makes it possible to detect the signals of single NPs as a whole, whereas a short dwell time causes the splitting of the NP signals between several single dwell time windows. The short dwell time led to several signals for every single particle, and, as a result, an overestimation of the number of TiO_2_ NPs and an underestimation of their sizes.

In the dwell time range of 16–20 ms, the number of TiO_2_ NPs for samples 4 and 5 is approximately similar, their distribution functions are similar, and the maximum of the distributions has a constant position (Appendix A). It allows us to consider the used experimental conditions optimal for determining the sizes of TiO_2_ NPs in these samples. For sample 3, a change in the distribution function was observed over the entire dwell time range (Appendix A); although the number of TiO_2_ NPs in the range of 10–16 ms was constant, the distribution function, position and number of NPs at the distribution maximum changed. This fact is probably caused by a higher content of TiO_2_ NPs in sample 3 compared to samples 4 and 5 (1745 particles versus 492 and 915 particles for samples 4 and 5, respectively, at 16 ms dwell time) (Table 1). For sample 3, with an increase in dwell time from 16 to 20 ms, the number of NPs at the distribution maximum decreased from about 200 to 100 particles, the right wing of the distribution became flatter, but the position of the distribution maximum remained constant (Appendix A). An increase in the number of high-intensity signals at the right wing of the distribution with a decrease in the total number of signals suggests that the selected conditions are not optimal for sample 3 due to the high concentration of TiO_2_ NPs. A long dwell time led to the time-unresolved detection of several NPs during several successive dwell time windows, i.e., an overestimation of the signal intensity of single NPs along with an underestimation of their number caused by the overlapping of NP fragments. For sample 3, the experiment was carried out under optimized conditions; the dilution factor was increased by 2 times for a better resolution of TiO_2_ NP signals in the time-resolved spectrum. The particle distribution functions of sample 3 also changed with increasing dwell time from 4 to 16 ms, had a constant position of the distribution maximum (156 nm) in the dwell time range of 16–20 ms, and a similar number of TiO_2_ NPs (808 and 774 particles) (Appendix A).

Thus, the signal distribution correction of the samples allows us to account for the contribution of the background signal to the size distribution of TiO_2_ NPs and choose the optimal conditions for their detection.

The shift of the right wing of the TiO_2_ NP size distribution after the correction was observed in a narrow range. For example, for sample 5 it was 206–216 nm versus 230–321 nm for distributions without background signal correction (Table 1). The TiO_2_ NP sizes at the right wing of the distributions after the correction did not exceed 218 nm. This was largely determined by the specifics of sample preparation, which included filtering the samples through a 0.22-μm syringe filter, and by the data processing algorithm—rare high-intensity signals with a frequency of fewer than 10 particles were not considered in the distribution.

Appendix A shows the signal distributions of control sample 6, which does not contain TiO_2_ NPs. Particle size distributions after correction contained single peaks of rare frequency presumably caused by possible cross-contamination due to the deposition of TiO_2_ NPs in the mass spectrometer input system. These data confirm the correctness of the calculation of the size distribution of TiO_2_ NPs in samples 3–5 after background signal correction (Appendix A).

It should be noted that in each case the developed algorithm of background signal correction provided the same LOD_size_ of TiO_2_ NPs of 71 nm (Appendix A). The LOD_size_ of the NPs mostly depends on the background level, below which it was impossible to distinguish the NP signal from the background signal. The proposed background signal correction eliminated most of the background from the sample distributions. After the background correction procedure, the LOD_size_ of TiO_2_ NPs depended on the sensitivity and settings of a particular device. In the conducted experiments, the signal intensity of 1 count after calibration corresponded to the TiO_2_ NP with a size of 71 nm. TiO_2_ NPs with a size smaller than 71 nm, if detected, were removed from the distribution, since the intensity of their signal (representing the summed signal of the background and NP) was in the range of the background signal fluctuations.

## 3. Materials and Methods

### 3.1. Samples

The objects of the study were freely available cosmetic products (Appendix A). According to the declared compositions of cosmetic products, samples 1 and 2 contained TiO_2_, and samples 3, 4, and 5 contained TiO_2_ NPs. Sample 6 did not contain TiO_2_ and was used as a control sample.

### 3.2. Materials and Reagents

For experimental studies, a non-ionic surfactant Triton X-100 (*t*-octylphenoxypolyethoxyethanol, Sigma-Aldrich, St. Louis, MO, USA), 0.22-μm syringe filters (Corning Incorporated, Berlin, Germany) and deionized water (NPC “Median Filter”, Moscow, Russia) were used. Standard reference dispersion (SRD) of 60 nm silver NPs (Sigma-Aldrich, USA) was used to determine the transport efficiency of ICP-MS. A 0.1 mg/mL dissolved Ti standard solution (Inorganic Ventures, Christiansburg, VA, USA) stabilized by high-purity nitric acid (Vecton, Novosibirsk, Russia) was used for calibration.

### 3.3. Instrumentation

The studies were performed on an iCAP RQ inductively coupled plasma mass spectrometer (Thermo Fisher Scientific, Waltham, MA, USA). The ICP-MS operating parameters were selected according to the manufacturer’s recommendations (Table 2). The signal was detected at *m*/*z* 48 for the isotope ^48^Ti with a natural abundance of 73.8%. The signal at *m*/*z* 107 for the isotope ^107^Ag with a natural abundance of 51.8% was also detected to determine the transport efficiency of ICP-MS. 

The presence of NPs in the cosmetic product samples was confirmed on the IG-1000 nanoparticle size analyzers (Shimadzu, Kyoto, Japan) at a voltage of 35 Vpp, a frequency of 500 kHz, and a time of 0.1 s.

The samples were sonicated in a GRAD 40–35 ultrasonic bath (Grad-Technology, Moscow, Russia).

### 3.4. SP-ICP-MS Instrument Setup

Due to the unavailability of reference monodisperse standards of TiO_2_ NPs, an indirect calibration procedure was used. Calibration of the SP-ICP-MS system for TiO_2_ NP size determination was performed according to the procedure described in [22,23].

The transport efficiency of ICP-MS was determined using the SRD of 60 nm silver NPs with the known number of NPs in the volume of dispersion. The number of NPs was calculated based on the Ag total mass concentration of the SRD and the mass of a single 60 nm silver NP calculated by Equation (6):(6)$$m_{\mathrm{NP}}=\;\frac{d^{3}\pi \rho }{6},$$
where *d* is the NP diameter, *ρ* is the NP density, $m_{\mathrm{NP}}$ is the mass of a single NP with a diameter *d* and density *ρ*.

The ratio between the number of NP peaks on the time-resolved spectrum and the number of NPs in the volume of standard dispersion introduced into the system allowed calculating the fraction of silver NPs that reached the detector considering the flow rate. The detection of the signal of the standard 60 nm silver NPs was performed under strictly optimized conditions, fulfilling the requirement to detect each single NP completely in a single dwell time [33]. The transport efficiency of the system with these operational parameters was 5.7% (Table 2). Further, it was assumed that (i) the calculated transport efficiency defined the part of titanium ions reaching the detector, (ii) the ionization efficiency in each case was equal to 100%, and (iii) the transport efficiency for silver and TiO_2_ NPs was the same.

The instrument calibration for the determination of TiO_2_ NP size was performed using Ti standard solution. Solutions for constructing the calibration curve were prepared before measurement. The mass of Ti (*m*) that reached the detector and generated a signal of known intensity was calculated based on the concentration of the standard solution and the transport efficiency according to Equation (7):(7)$$m=\eta_{\mathrm{TE}}Vt_{\mathrm{dt}}C,$$
where $\eta_{\mathrm{TE}}$ is the transport efficiency, *V* is the sample flow rate, $t_{\mathrm{dt}}$ is the dwell time, *C* is the Ti concentration in the standard solution.

The transformed calibration curve of the dependence of the signal intensity on the real detected mass of Ti (*m*) was used in the determination of TiO_2_ NPs assuming that the behavior of the Ti ionic form of the Ti standard solution and TiO_2_ NPs was the same. The mass of Ti (*m*) corresponding to the signal intensity was recalculated into the mass of TiO_2_ NPs ($m_{\mathrm{NP}}$) considering the TiO_2_/Ti molar ratio equal to 1.67. The size of TiO_2_ NP (*d*) was calculated by Equation (8) using the NP mass ($m_{\mathrm{NP}}$) and density (ρ equal to 4.23 g/cm^3^ for TiO_2_) and assuming the spherical geometry of the particle:(8)$$d=\sqrt[{3}]{\frac{6m_{\mathrm{NP}}}{\pi \rho }}.$$

### 3.5. Sample Preparation of Cosmetic Product Samples

Preparation of the studied samples of cosmetic products for analysis was performed according to a well-known procedure [19]. A 0.05-g homogenized cosmetic product sample was placed in a 50-mL tube, then 0.1% Triton X-100 surfactant (*v*/*v*) was added, and the solution was made up to the mark with deionized water. The mixture was shaken and sonicated for 30 min. Large particles and agglomerates of TiO_2_ NPs were filtered by a 0.22-μm syringe filter. The filtrate was diluted with deionized water and sonicated for 5 min to evenly distribute the NPs in the solution. All measurements were performed immediately after the suspension preparation to prevent agglomeration of TiO_2_ NPs in time.

### 3.6. Data Processing for SP-ICP-MS

The obtained raw data were processed using Microsoft Office Excel software. The signal intensities from the time-resolved spectrum were represented by graphical distributions in «frequency–intensity» coordinates, where frequency is the number of signals of a definite intensity. The change in signal intensity in a wide range of dwell times was estimated in absolute unit–counts. The ratio of counts per second (cps) was recalculated considering dwell time. The absolute intensity value (counts) was used to represent signal distributions. Moreover, the capabilities of the SPCal software were used to calculate the background signal [32].

## 4. Conclusions

We considered the specifics of the millisecond time-resolved SP-ICP-MS determination of TiO_2_ NP sizes in cosmetic products. The analysis of TiO_2_ NPs in cosmetic products is a difficult task due to the high background signal and the requirements for optimal detection conditions by this method. It was found that the background signal in the analysis of TiO_2_ NPs in cosmetic product samples at the millisecond time resolution was determined by the background content of Ti in its dissolved form and interferents in deionized water, and the background signal of the cosmetic matrix can be regulated by the sample dilution.

The background signal was shown to make a significant contribution to the distribution of sample signals and depend on the detection conditions. Overlapping of the distributions of background signals and TiO_2_ NPs complicates the correct estimation of the NP signals. Using a σ-criteria for correcting the background signal leaves either a false positive background signal or accepts low-frequency signals of NPs as false negatives in the overlapped area. An alternative correction procedure for the frequency and intensity of the background signal was proposed, which allows the separating of the background and TiO_2_ NP signals in the overlapped area.

The distributions of TiO_2_ NP signals in real cosmetic product samples were studied in the dwell time range of 4–20 ms. The limit of detection of the size (LOD_size_) of TiO_2_ NPs was 71 nm when the developed procedure for correcting the background signal was used. It has been shown that with SP-ICP-MS determination of the TiO_2_ NP sizes in real cosmetic products, the LOD_size_ does not depend on the background signal and is determined by the ICP-mass spectrometer sensitivity.

## Figures and Tables

**Figure 1 molecules-27-07748-f001:**
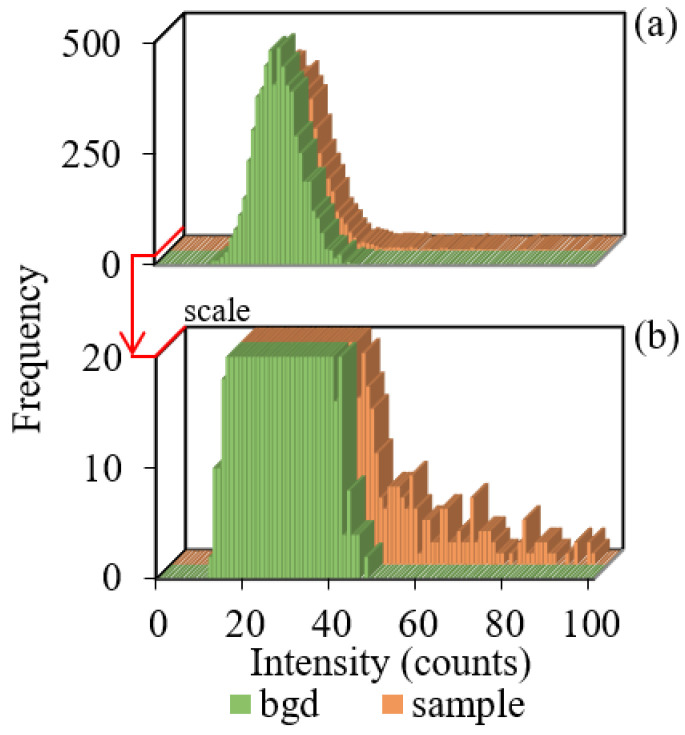
Signal distributions of the background and sample 3 containing TiO_2_ NPs in (**a**) full and (**b**) enlarged scales.

**Figure 2 molecules-27-07748-f002:**
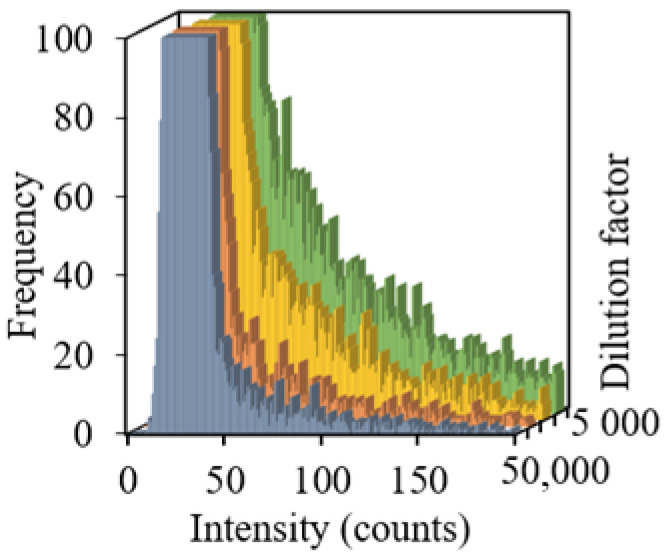
Signal distributions of sample 5 at dilution factors from 5000 to 50,000.

**Figure 3 molecules-27-07748-f003:**
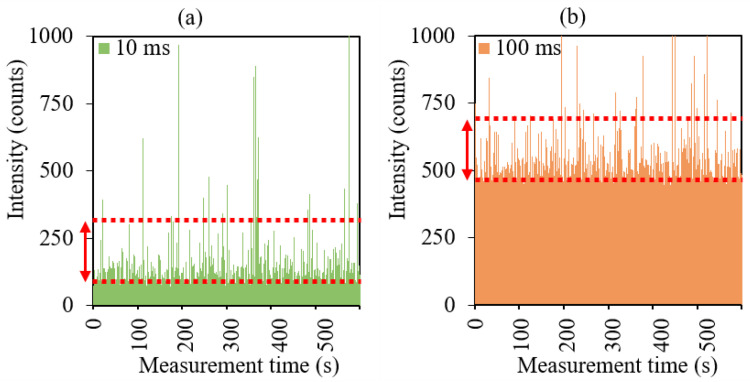
Time-resolved spectra at dwell times of (**a**) 10 and (**b**) 100 ms; (**c**) signal distributions of sample 3.

**Figure 4 molecules-27-07748-f004:**
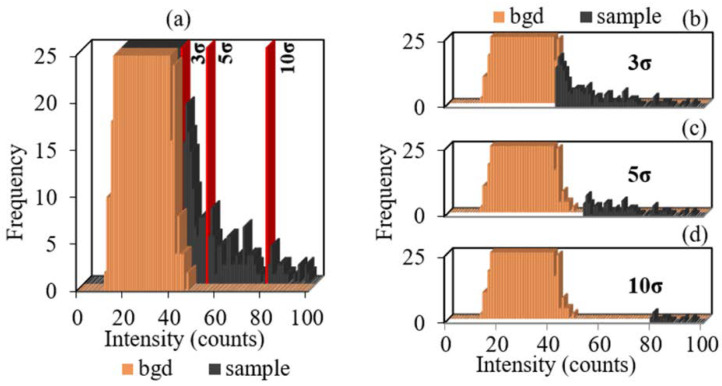
The signal distributions of (**a**) the background and sample 3 and particle distributions after correction by (**b**) 3σ, (**c**) 5σ, and (**d**) 10σ threshold.

**Figure 5 molecules-27-07748-f005:**
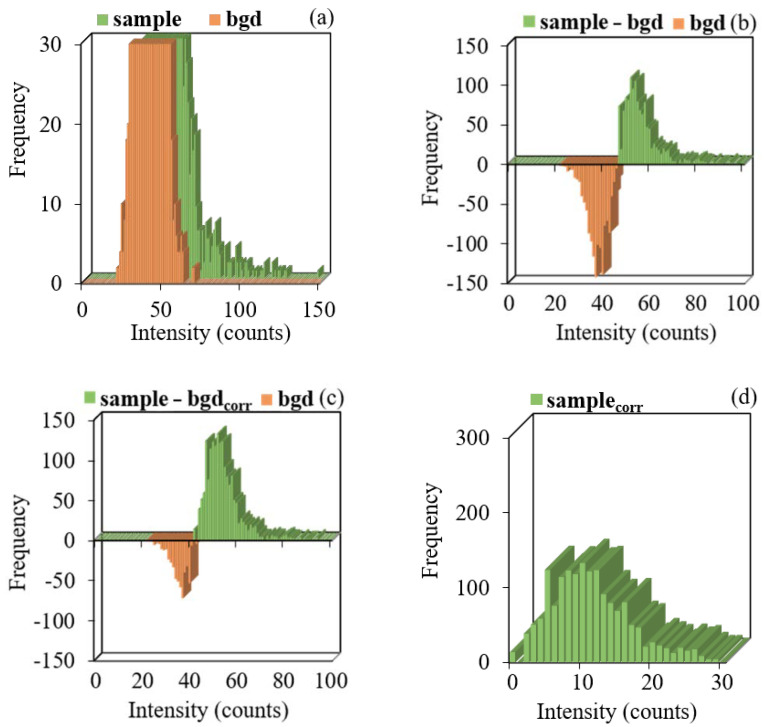
Signal distributions of the background and sample 3: (**a**) before correction of the background signal, after correction, (**b**) for the frequency of the background signal; corrected frequency of the background signal considering the number of NPs in the sample, (**c**) and considering the mean intensity of the background signal (**d**).

**Table 1 molecules-27-07748-t001:** Change in characteristics of TiO_2_ NP size distributions of cosmetic product samples after background signal correction.

**Sample 3**	**Dwell Time (ms)**		**4**	**10**	**16**	**20**
	Maximum distribution (nm)	Before	166	217	257	276
		After	112	141	152	152
	Right-wing distribution (nm)	Before	226	260	291	308
		After	194	200	212	218
	Number of particles	Before	-	-	-	-
		After	2893	1763	1745	1378
**Sample 4**	**Dwell Time (ms)**		**4**	**10**	**16**	**20**
	Maximum distribution (nm)	Before	146	200	230	248
		After	122	128	146	146
	Right-wing distribution (nm)	Before	216	238	262	286
		After	194	192	194	188
	Number of particles	Before	-	-	-	-
		After	1863	1204	492	467
**Sample 5**	**Dwell Time (ms)**		**4**	**10**	**16**	**20**
	Maximum distribution (nm)	Before	162	224	260	282
		After	122	128	136	140
	Right-wing distribution (nm)	Before	230	276	299	321
		After	206	214	216	216
	Number of particles	Before	-	-	-	-
		After	1455	1095	915	873

**Table 2 molecules-27-07748-t002:** Change in characteristics of TiO_2_ NP size distributions of cosmetic product samples after background signal correction.

Parameter	Value	
**Operating parameters of the iCAP RQ mass spectrometer**		
RF power (W)	1550	
Plasma gas flow (L/min)	14	
Auxiliary gas flow (L/min)	0.8	
Nebulizer gas flow (L/min)	0.7	
Sample flow rate (mL/min)	0.28	
Analysis mode	standard	time resolution
Dwell time (ms)	10	0.1–20, 100
Total measurement time (s)	-	60
Mass (amu)	48	48, 107
**Parameters for SP-ICP-MS determination of TiO_2_ NP size**		
Ionization efficiency (%)	100	
Transport efficiency (%)	5.7	
Molar ratio TiO_2_/Ti	1.67	
TiO_2_ density (g/cm^3^)	4.23	

## Data Availability

Not applicable.

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
