# Peer review of "A Novel Method for the Background Signal Correction in SP-ICP-MS Analysis of the Sizes of Titanium Dioxide Nanoparticles in Cosmetic Samples"

_molecules, 2022, doi:10.3390/molecules27227748_

Round 1

Reviewer 1 Report

The authors have been developed an approach for analysis of sizes of titanium dioxide nanoparticles. An emerging technique SP-ICP-MS was used for this purpose. The manuscript can be recommended for publication after revision.

- The abstract should be revised with emphasis on the novelty of the work.

- An open-source Python-based SP/SC ICP-MS processing platform (https://doi.org/10.1039/D1JA00297J) must be mentioned and compared with the proposed correction procedure.

- The results obtained by alternative methods are highly recommended to be added to the experimental part.

Author Response

We express our deep appreciation to the editor and reviewers for the professional discussion of the research results and the comments made that improved the content of our article.

Point 1: The abstract should be revised with emphasis on the novelty of the work.

Response 1: We agree with the remark. The abstract has been finalized taking into account the novelty.

Point 2: An open-source Python-based SP/SC ICP-MS processing platform (https://doi.org/10.1039/D1JA00297J) must be mentioned and compared with the proposed correction procedure.

Response 2: We agree with the remark. The article has been updated. The SPCal platform is discussed in the introduction, experimental data are also processed using this platform in Section 2.3.

Point 3: The results obtained by alternative methods are highly recommended to be added to the experimental part.

Response 3: We agree with the remark. In Section 2.2, the results of the study of samples by dynamic light scattering using the IG-1000 nanoparticle size analyzer have been added.

Reviewer 2 Report

To check the accuracy of the results of determining the sizes of NPs by the proposed method, it was necessary to analyze the CRM. Or it was necessary to analyze samples of cosmetics by another method (for example, electron microscopy).

Author Response

We express our deep appreciation to the editor and reviewers for the professional discussion of the research results and the comments made that improved the content of our article.

Point 1: To check the accuracy of the results of determining the sizes of NPs by the proposed method, it was necessary to analyze the CRM. Or it was necessary to analyze samples of cosmetics by another method (for example, electron microscopy).

Response 1: We agree with the remark. In Section 2.2, the results of the study of samples by an alternative method - dynamic light scattering, obtained on the IG-1000 nanoparticle size analyzer, have been added.

Round 2

Reviewer 1 Report

The paper can be accepted without any further changes.